# PRMT5-mediated methylation of STAT3 is required for lung cancer stem cell maintenance and tumour growth
Yoshinori Abe [1,2,4], Takumi Sano[2,4], Naoki Otsuka[2], Masashi Ogawa[2] & Nobuyuki Tanaka [2,3] ✉

STAT3 is constitutively activated in many cancer types, including lung cancer, and can induce cancer cell proliferation and cancer stem cell (CSC) maintenance. STAT3 is activated by tyrosine kinases, such as JAK and SRC, but the mechanism by which STAT3 maintains its activated state in cancer cells remains unclear. Here, we show that PRMT5 directly methylates STAT3 and enhances its activated tyrosine phosphorylation in non-small cell lung cancer (NSCLC) cells. PRMT5 expression is also induced by STAT3, suggesting the presence of a positive feedback loop in cancer cells. Furthermore, methylation of STAT3 at arginine 609 by PRMT5 is important for its transcriptional activity and support of tumour growth and CSC maintenance. Indeed, NSCLC cells expressing the STAT3 mutant which R609 was replaced to alanine (R609K) show significantly impaired tumour growth in nude mice. Overall, our study reveals a mechanism by which STAT3 remains activated in NSCLC and provides a new target for cancer therapeutic approaches.

The signal transducer and activator of transcription (STAT) protein family members were originally identified as regulatory transcription factors involved in the cellular response to interferons (IFNs)[1]. Subsequently, STATs were found to be activated by various cytokines, cell growth factors, and hormones via the intracellular, non-receptor tyrosine kinases, Janus kinases (JAKs), to regulate cellular responses to these ligands[1,2]. STAT family proteins include STAT1–4, 5 A, 5B, and 6. After being phosphorylated at tyrosine residues by JAKs, STATs can dimerize and translocate into the nucleus to regulate target gene expression in response to extracellular signals and control various cellular responses, such as immune system activities and cell growth[3–5]. In cancer cells, STAT proteins are activated by increased expression of cytokines/growth factors and by aberrant activation of their signalling molecules. STATs are involved in tumour cell growth, resisting cell death, and cancer stem cell (CSC) maintenance via expression of their target genes[3,5]. The involvement of STAT family proteins in tumorigenesis and cancer progression has been widely reported, particularly for STAT3 and STAT5, which are constitutively activated in many cancer cells and have been experimentally shown to be important in cancer progression[5,6].

STAT3 was identified as a transcription factor that binds to interleukin-6 (IL-6)-responsive elements in the promoters of genes encoding acute-phase proteins[7]. The IL-6/JAK/STAT3 signalling pathway plays a pro-inflammatory role by promoting the process of epithelial–mesenchymal transition and many cellular responses, including immune responses, angiogenesis, and cell growth and differentiation[4,8,9]. Furthermore, STAT3 is activated by various inflammation-related and immune regulatory cytokines and cell growth factors that have significant functions in wound healing and tissue repair. Therefore, excessive activation of STAT3 in cancer cells, as well as in cells in the inflammatory tumour microenvironment (TME), can cause an inflammation-driven repair response[10]. This is defined as the "wounds that do not heal" model of cancer, which can explain tumour promotion and progression[11,12]. In cultured cells, a constitutively activated STAT3 mutant was shown to function as an oncogene[13], suggesting that STAT3 itself can reprogram normal cells into CSCs[14,15]. In this context, STAT3 can help generate CSCs by inducing the expression of reprogramming factors[16]. Moreover, the inflammatory TME may promote the establishment and maintenance of CSCs via STAT3[17,18]. In addition to these functions, STAT3 can induce metabolic reprogramming, a hallmark of cancer, by regulating various metabolic processes in cancer cells, including aerobic glycolysis, oxidative phosphorylation, glutamine meta-bolism, lipid synthesis, and lipid catabolism[3]. Therefore, the inhibition of STAT3 activity is expected to inhibit cancer cell proliferation and suppress CSCs, but clinically effective drugs that exploit this mechanism are not yet available[10].

STAT3 activation is initiated by phosphorylation of a tyrosine residue (Y705) by JAK family kinases, JAK1, JAK2, JAK3, and TYK2. This tyrosine residue can also be phosphorylated by non-receptor kinases, such as SRC

[1]Laboratory of Molecular Analysis, Nippon Medical School, Tokyo, Japan. [2]Department of Molecular Oncology, Institute for Advanced Medical Sciences, Nippon Medical School, Tokyo, Japan. [3]Division of Cell Physiology, Department of Physiology and Cell Biology, Graduate School of Medicine, Kobe University, Kobe, Japan. [4]These authors contributed equally: Yoshinori Abe, Takumi Sano. ✉e-mail: nobuta@nms.ac.jp

and ABL[2,4]. Tyrosine-phosphorylated STAT3 forms dimers that translocate to the nucleus to activate the expression of target genes. STAT3 is also phosphorylated at the serine 727 (S727) residue by mitogen-activated protein kinase (MAPK) protein family members, such as extracellular signal-regulated kinase (ERK)1, ERK2, p38, and c-Jun N-terminal kinase (JNK)[19,20]. In mutant STAT3-expressing cells in which S727 was replaced with alanine, the DNA binding activity of STAT3 was not affected, but the transcriptional activity was decreased, suggesting that this phosphorylation is required for maximal STAT3 activity. Moreover, the expression of this mutant STAT3 in mice prevented the lethality associated with STAT3 deficiency, but its transcriptional activity was reduced by half[21]. Additionally, acetylation and methylation of STAT3 can reportedly enhance its transcriptional activity[19,20].

Constitutive activation of STAT3 has been reported in many cancer types. However, STAT3 gain-of-function mutations have been observed in some leukaemias, but less frequently in solid tumours[22]. Because STAT3 can induce the expression of cytokines/growth factors, and their receptors activate STAT3, it is possible that STAT3 is permanently activated by a positive feedback loop in the TME[5]. In JAK/STAT regulation systems, the activation of STAT3 by JAK is inhibited by suppressor of cytokine signalling 3 (SOCS3), which is transcriptionally induced by STAT3[23]. Thus, given the importance of STAT3 for cancer cell survival[5,24], the mechanism of constitutive STAT3 activation in cancer cells needs to be fully elucidated.

Lung cancer is a leading cause of cancer-related death, with more than 85% of lung cancer cases being classified as non-small cell lung cancer (NSCLC)[25]. One of the major oncogenic drivers in NSCLC is mutated epidermal growth factor receptor (EGFR) that results in constitutive activation. STAT3 is one of the key signalling mediators downstream of EGFR, and STAT3 is indeed constitutively activated in about 50% of NSCLC tissues and cell lines[26–28]. In addition to this pathway, the HEDGEHOG (HH) pathway is known to be activated in NSCLC[29]. Experimental evidence suggests that STAT3 activation is enhanced downstream of HH signalling[30]. The HH signalling pathway plays a central role in the growth, patterning, and morphogenesis of organisms, and is also involved in adult organ homeostasis, stem cell maintenance, and cancer development[31–33]. We previously found that HH signalling in cancer cells can induce the expression of protein arginine methyltransferase 5 (PRMT5), with arginine methylation by PRMT5 being important for activation of the effector transcription factor GLI1 protein[34]. PRMTs are enzymes that can methylate

arginine residues in certain proteins and have been reported to regulate gene expression, signal transduction, and mRNA splicing fidelity[35]. Of the nine members of the PRMT protein family, PRMT5 is believed to be involved in cancer through various functions, including tumour growth[36], stem cell reprogramming and maintenance[37], and immune evasion[38]. Furthermore, overexpression of PRMT5 has been observed in a variety of cancer types[39]. In the present study, we found that PRMT5-mediated methylation is important for constitutive activation of STAT3, as well as for CSC maintenance and tumorigenesis in lung cancer cells. These results suggest that targeting the arginine methylation of STAT3 by PRMT5 is a promising new approach for treating lung cancer.

## Results

### *PRMT5* mRNA expression levels are increased in lung cancer cells

We first analysed PRMT5 expression levels in NSCLC samples. As shown in Fig. 1a, b, *PRMT5* mRNA expression levels were enhanced in the two most common NSCLC subtypes, lung adenocarcinoma and lung squamous cell carcinoma, as determined using RNA sequencing (RNA-seq) transcriptomic data from The Cancer Genome Atlas (TCGA) database[40]. *PRMT5* expression level and copy number were correlated, with higher *PRMT5* expression being associated with worse overall survival in lung adenocarcinoma patients (Supplementary Fig. 1a, b). Moreover, mRNA expression of methylosome protein 50 (MEP50), an adaptor molecule between PRMT5 and its substrates[41,42], was also enhanced (Supplementary Fig. 1c, d). This suggests that methylosomes containing PRMT5 function more potently in NSCLC. *PRMT5* expression levels were higher in NSCLC, which exhibits enhanced signalling of HH, a known PRMT5 inducer[34], than in conditions such as oesophagogastric, colorectal, prostate, and pancreatic cancers[43–46] (Supplementary Fig. 1e). The ranking of *PRMT5* expression in all 1,000 cancer cell lines using the Cancer Cell Line Encyclopedia (CCLE; https://sites.broadinstitute.org/ccle/) also showed high *PRMT5* expression levels in NSCLC cells (Supplementary Fig. 1f). These results suggest that methylation by PRMT5 plays an important role in NSCLC, including in cancer progression. Furthermore, using the CCLE, we also found that PRMT5 protein (Fig. 1c) and mRNA (Supplementary Fig. 1g) expression levels correlated with STAT3 protein expression levels in 12 and 8 NSCLC cell lines, respectively, suggesting that STAT3 is involved in PRMT5 expression.

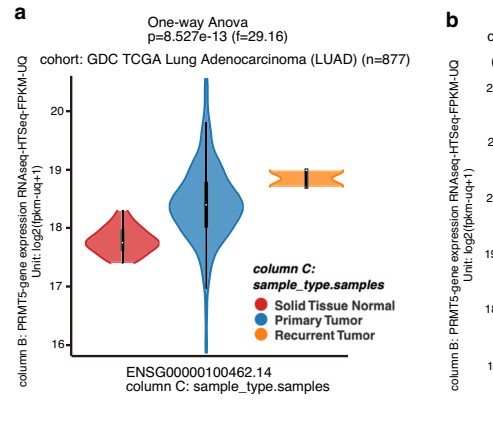
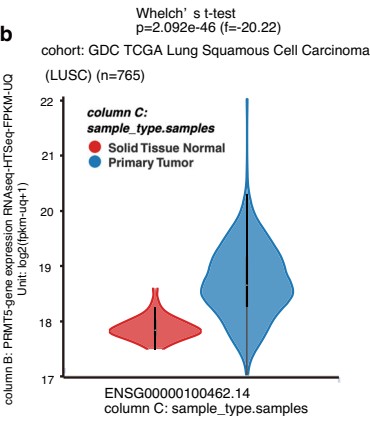
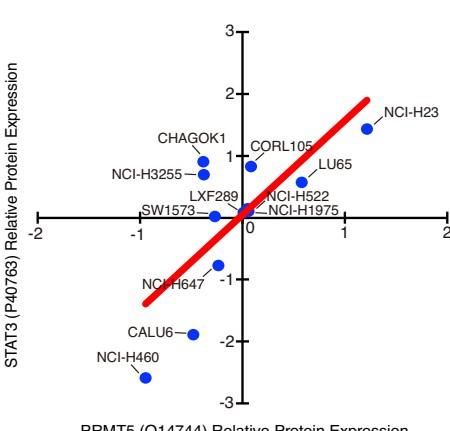

**Fig. 1 | *PRMT5* expression levels are increased in lung cancer cells. a, b** *PRMT5* expression is elevated in **a** lung adenocarcinoma and **b** lung squamous cell carcinoma. RNA-seq data from The Cancer Genome Atlas (TCGA) were analysed on the UCSC Xena browser. **c** Correlation between STAT3 and PRMT5 protein levels in 12 NSCLC cell lines (NCI-H23, LU65, CORL105, CHAGOK1, NCI-H3255, NCI- H522, NCI-H1975, LXF289, SW1573, NCI-H647, CALU6, NCI-H460). The data were obtained from the Cancer Cell Line Encyclopedia (CCLE) and obtained expression data were analysed using Data Explorer at the depmap portal (https:// depmap.org/portal/).

## STAT3 activates *PRMT5* gene transcription

Because *PRMT5* mRNA expression levels were increased in NSCLC samples, we next analysed the relationship between the transcription factor STAT3, which is constitutively activated in NSCLC, and *PRMT5* expression. Using short hairpin RNA (shRNA) to knock down *STAT3* expression (shSTAT3), we observed reduced PRMT5 protein expression levels in normally growing murine diploid NIH3T3 cells. STAT3 expression was induced by v-SRC, a typical oncogenic signalling molecule known to activate STAT3[47] (Supplementary Fig. 2a), suggesting that STAT3 is involved in the regulation of PRMT5 expression in cancer cells. Moreover, *PRMT5* mRNA expression was upregulated by the STAT3 activator IL-6, while this induction was suppressed by STAT knockdown (Supplementary Fig. 2b). EGF-induced *PRMT5* mRNA expression was also inhibited by the STAT3 inhibitor LLL-12 (Supplementary Fig. 2c), suggesting that *PRMT5* gene transcription is activated by STAT3. To further analyse the induction of PRMT5 expression by STAT3 in NSCLC cells, we generated STAT3-knockout (STAT3-KO) cells by using the CRISPR/Cas9 system in A549 (EGFR wild type, adenocarcinoma) and HCC827 cells (EGFR mutant, adenocarcinoma). As expected, the induction of *PRMT5* mRNA expression by IL-6 was markedly suppressed by STAT3 KO in A549 and HCC827 cells (Fig. 2a). Moreover, PRMT5 protein expression and induction by IL-6 were also suppressed by STAT3 KO (Fig. 2b), suggesting that PRMT5 is induced by STAT3 in NSCLC. Furthermore, the *PRMT5* promoter, which includes putative STAT3 binding sites[48], was activated by a constitutively active mutant of STAT3 (STAT3C)[13] (Fig. 2c–e). As expected, we found that STAT3 directly bound to the regulatory regions of *PRMT5* in NSCLC cells (Fig. 2f, g [A549], Fig. 2h, i [HCC827]). Binding of STAT3 to the *PRMT5* gene locus in NSCLC cells was also confirmed by ChIP-Seq analysis in the UCSC Genome Browser database (Supplementary Fig. 2d).

## PRMT5 induces tumour growth and CSC maintenance in NSCLC cells

We next analysed the role of PRMT5 in HCC827 and A549 NSCLC cells. The proliferation rates of cells with knockdown of PRMT5 (Fig. 3a) and its adaptor molecule MEP50 (Fig. 3b) were slightly reduced compared with those of the parental cells (Fig. 3c, d). Cancer arises from CSCs, also known as cancer-initiating cells, which have tumorigenic potential, self-renewal properties, and long-term tumour repopulating activity[14,49,50]. It has been shown that CSCs can form spheres under low-attachment culture conditions in media containing growth factors[51,52]. Indeed, these cells formed primary spheres and secondary spheres, characteristic of stem cells[53], but PRMT5 knockdown reduced their number (Fig. 3e, f). Certain reprogramming factors, such as SOX2, OCT3/4, and NANOG, have exhibited pluripotent stem cell (iPSC)-inducing activity and are a hallmark of CSCs[15]. In sphere-forming cells, the expression levels of these reprogramming factors were elevated (Supplementary Fig. 3a, b). In addition, the population of highly activated aldehyde dehydrogenase (ALDH) cells, which is a hallmark of lung CSCs, was reduced in PRMT5-knockdown cells (Fig. 3g, h, Supplementary Fig. 2c). These results suggest that PRMT5-mediated methylation is involved in CSC maintenance. Like the decreased number of sphere-forming cells, the growth of A549 cell tumours transplanted into nude mice was also reduced in PRMT5-inducible-knockdown cells (Fig. 3i, j, Supplementary Fig. 3d–f).

## STAT3 activity is upregulated by PRMT5-mediated methylation

Next, we analysed the functional effects of the interaction between PRMT5 and STAT3 in cancer cells. The number of sphere-forming cells among HCC827 and A549 cells was increased by IL-6 stimulation, but this increase did not occur in PRMT5-knockdown cells (Fig. 4a, b). Related to this, the induction of MCP-1 mRNA, which is known to be regulated by the IL-6–STAT3 pathway[54], was suppressed by PRMT5 knockdown (Supplementary Fig. 4a, b). Furthermore, PRMT5 knockdown also suppressed the increase in the number of sphere-forming cells following hepatocyte growth factor (HGF) treatment, which is known to induce oncogenic transformation via the MET–STAT3 pathway[55] (Supplementary Fig. 3c, d). These

results suggest that STAT3 is involved in CSC maintenance downstream of PRMT5. Indeed, IL-6-induced Y705 phosphorylation was suppressed in PRMT5-knockdown cells (Fig. 4c), as was STAT3 transcriptional activity (Fig. 4d, e). However, PRMT5 knockdown did not affect the transcriptional activity of transcription factors NF-κB and HIF1[56], which are known to be activated in other cancer types (Supplementary Fig. 4e, f).

Candidate arginine methylation sites in STAT3 were predicted by PRmePRed (Fig. 4f, Supplementary Fig. 4g). To analyse whether STAT3 protein is methylated by PRMT5, we performed an in vitro methylation assay. Myc-tagged wild-type STAT3 (STAT3 WT) and its mutants purified by immunoprecipitation were methylated by recombinant human Flag-tagged PRMT5/His-tagged MEP50 complex (Fig. 4g, h). In contrast, upon amino acid substitution (arginine to lysine) at candidate methylation sites of STAT3, methylation was clearly suppressed by arginine 609 (R609K) mutation and weakly suppressed by arginine 518 (R518K) mutation (Fig. 4g, h). Next, wild-type STAT3 and STAT3 mutants were transiently expressed in the STAT3-KO normal human bronchial epithelial cell line BEAS2B (Supplementary Fig. 4h). As shown in Fig. 4i, methylated proteins could be detected at the same position as STAT3, and the degree of methylation was similar to that in the in vitro methylation assay. In addition, R609K mutant STAT3 lost most of its transcriptional activity in STAT3-KO BEAS2B cells (Fig. 4j). We also found that PRMT5 forms a complex with STAT3 via MEP50 (Fig. 4k). Furthermore, we showed that STAT3 and PRMT5 were colocalized in the cytoplasm, but only STAT3 was localized in the nucleus (Supplementary Fig. 4i, j). These results suggest that STAT3 is methylated in the methylosome complex formed by PRMT5/MEP50[41,42] in the cytoplasm, resulting in STAT3 activation, translocation into the nucleus, and target gene activation.

## PRMT5-mediated STAT3 methylation is required for tumorigenesis and CSC maintenance in NSCLC cells

Previously, it has been shown that STAT3 plays an important role in v-SRC transformation, as dominant negative STAT3 mutants can inhibit v-SRC transformation[57]. As shown in Supplementary Fig. 5a–f, tumour suppressor p53-deficient embryonic fibroblasts expressing v-SRC showed foci formation[58], a hallmark of cancer cells. However, this ability was reduced by PRMT5, MEP50, or STAT3 knockdown, suggesting that the PRMT5/MEP50/STAT3 pathway is important for CSC generation. Therefore, we examined the impact of STAT3 R609K on tumour growth and CSC maintenance. Active phosphorylation of STAT3 Y705 was inhibited in the STAT3-R609K-expressing STAT3-KO HCC827 cells (Fig. 5a), and nuclear translocation of STAT3 by IL-6 was also suppressed (Fig. 5b). Moreover, ChIP-qPCR analysis revealed that STAT3 R609K mutant did not directly bind to the regulatory region of *PRMT5* (Fig. 5c). As shown in Fig. 5d, STAT3-KO HCC827 cells showed slightly decreased cell proliferation rates, but these were increased to levels similar to those of the parental cells after wild-type STAT3 was expressed. However, STAT3-R609K mutant-expressing cells did not show recovered proliferation rates (Fig. 5d). Moreover, STAT3-KO cells had a markedly decreased number of sphere-forming cells, and the R609K mutant showed reduced recovery of sphere-forming cells compared with wild-type STAT3 (Fig. 5e). Furthermore, tumour growth in nude mice was markedly reduced in STAT3-KO cells and was restored by wild-type STAT3 expression, similar to the level in parental cells, while this recovery was not observed in mutant STAT3-R609K-expressing cells (Fig. 5f, g and Supplementary Fig. 6). These results suggest that methylation of STAT3 R609 by PRMT5 is important for CSC maintenance and tumour growth in NSCLC.

## Discussion

This study reveals that PRMT5 can methylate and activate STAT3, which can then induce PRMT5 expression, thereby maintaining both in an activated state (Fig. 5h). These results reveal a previously unknown mechanism of STAT3 and PRMT5 activation. Furthermore, we found that activation of this PRMT5–STAT3 pathway is important for tumour growth and CSC maintenance in NSCLC. The molecular mechanism of the reprogramming

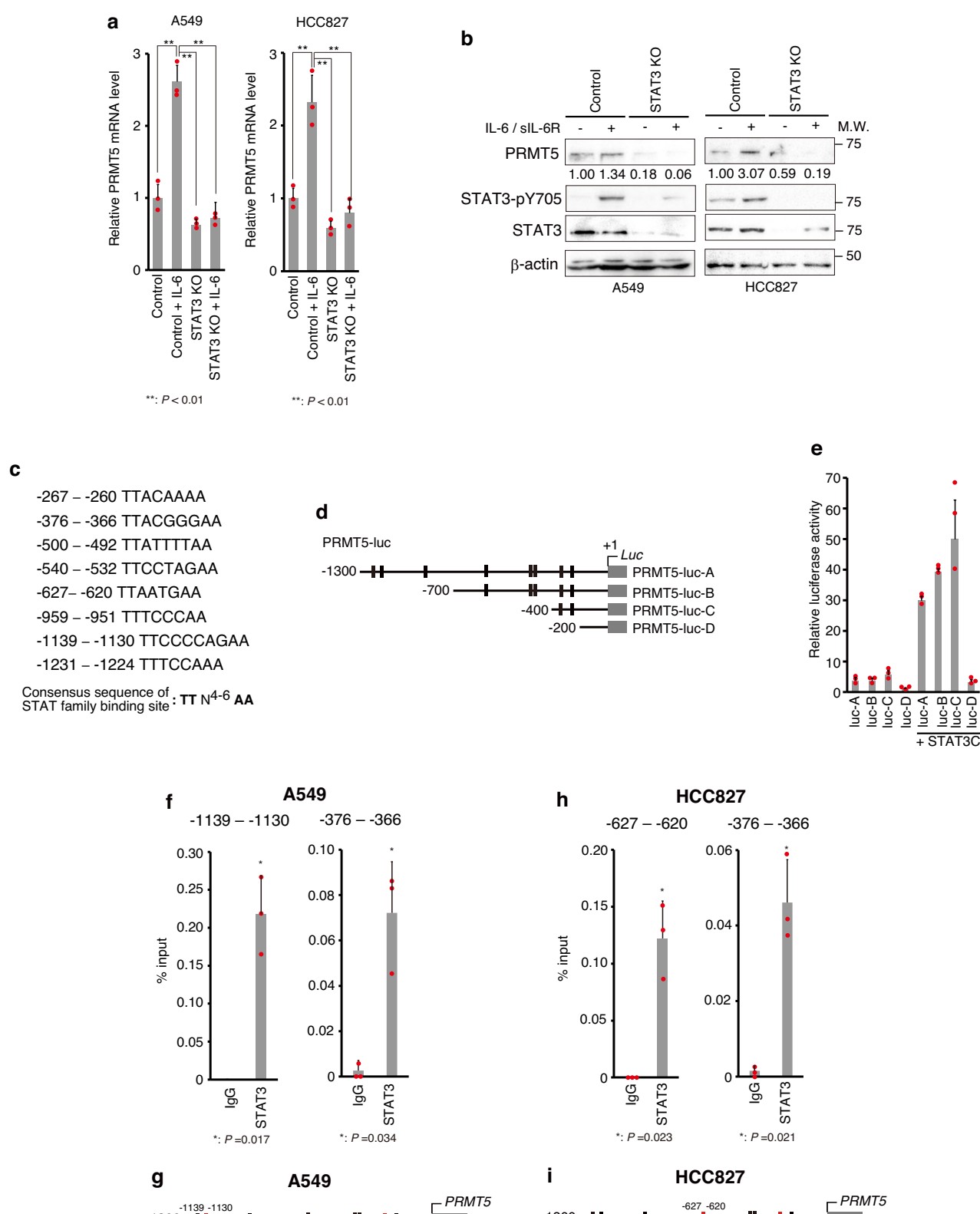

process, in which normally differentiated cells are transformed into CSCs, is thought to involve regulators and control mechanisms similar to those of iPSCs[15]. Indeed, the reprogramming transcription factors and chromatin modifiers required for iPSC establishment have been shown to be associated with cancer development[15,59]. Because cancer is caused by stepwise changes to oncogene and tumour suppressor gene expression and/or protein

activity[60,61], cancer-promoting signals occurring as a result of these changes can induce cell reprogramming. In cultured cells, constitutively activated STAT3 mutant has been shown to function as an oncogene[13], suggesting that STAT3 itself can reprogram normal cells into CSCs[14,15]. Indeed, it has been shown that STAT3 is involved in CSC generation by inducing the expression of reprogramming factors such as SOX2 and OCT4[16]. In the

**Fig. 2 | STAT3 induces PRMT5 expression. a, b** Recombinant interleukin (IL)-6-mediated STAT3 activation enhances PRMT5 expression in NSCLC cells. Cells were treated with 50 ng/ml recombinant IL-6 and 50 ng/ml soluble IL-6 receptor (sIL-6R) for 3 h before harvesting. In **a**, *PRMT5* expression levels were analysed by qPCR. Results are presented as the mean ± standard deviation (SD) from three experiments. Tukey's honestly significant difference test was applied for statistical comparisons. In **b**, PRMT5 or STAT3 protein levels were determined by western blot analysis using antibodies for PRMT5 (abcam; ab10941) or STAT3 (CST; 12640). PRMT5 expression levels were quantified with normalization to the β-actin expression intensity. STAT3 activation was monitored by western blotting using an anti-phosphorylated STAT3 (Y705) antibody (CST; 9138). STAT3. **c** Putative STAT3 binding sequences in the *PRMT5* promoter regions. **d, e** STAT3 is involved in *PRMT5* expression. **d)** Schematic illustration of the reporter plasmid (PRMT5-luc) constructs. The vertical line denotes the predicted STAT3 binding site. **e** STAT3-mediated upregulation of *PRMT5* expression was analysed by luciferase assays using the reporter plasmids shown in **c**. The results are expressed as firefly/*Renilla* ratios and are presented as the mean ± SD from three experiments. **f–i** STAT3 directly binds to the regulatory region of *PRMT5* in NSCLC cells. STAT3 binding sites were assessed by ChIP assays using A549 cells **f** or HCC827 cells **h**. Results are presented as the mean ± SD from three experiments. Unpaired two-tailed Student's t-test was applied for statistical comparisons. **g, i** Schematic illustration of the STAT3 binding sites in A549 cells **g** or HCC827 cells **i**, shown as red vertical lines. Representative results of **a**, **b**, and **e** from two independent experiments are shown. Source data are provided in Supplementary Data 1. Full immunoblot and electrophoresis images are shown in Supplementary Fig. 7.

present study, we found that STAT3 knockdown suppressed fibroblast transformation by v-SRC and reduced the number of CSCs in NSCLC. These results suggest that STAT3 is important for CSC generation and maintenance, especially in NSCLC tumours. Thus, the mechanism of constitutive STAT3 activation may be important for CSC development, dedifferentiation of differentiated cancer cells into CSCs, and CSC maintenance. Furthermore, because STAT3 KO also reduces cell proliferation rates, these results suggest that the induction of STAT3 by PRMT5 is involved in tumorigenesis by regulating both stemness and proliferation.

In this study, we found that PRMT5 can activate STAT3 via arginine methylation. In previous work, PRMT5 knockdown was shown to suppress cell proliferation, while PRMT5 overexpression caused increased cancer cell proliferation[36,62], suggesting a role in cancer cell growth. Moreover, PRMT5 is also important for maintaining pluripotency of mouse embryonic stem cells[63]. Additionally, the induction of iPSCs by Yamanaka factors (OCT4, SOX2, KLF4, and C-MYC) is reduced in cells lacking PRMT5[64]. Thus, PRMT5 is probably important for CSC generation and maintenance. We previously showed that GLI1, the effector transcription factor in the HEDGEHOG pathway that is thought to be important in CSC development, is activated by PRMT5/MEP50[34]. It has been proposed that HH signalling maintains the stemness of CSCs[65,66], suggesting that PRMT5/MEP50 are also important for CSC maintenance. In the present study, we also found that methylation of STAT3 by PRMT5/MEP50 in NSCLC cells is important for its transcriptional activity, for the generation and maintenance of CSCs, and for tumour growth in mice. Therefore, although the dependence may vary by cancer type, our data demonstrate that the PRMT5/MEP50/STAT3 pathway is important for CSCs in NSCLC.

Activation of STAT3 by JAK leads to transcriptional induction of the JAK inhibitor SOCS3 by STAT3[23], suggesting that STAT3 activity can oscillate in JAK-activated cancer cells and EGFR-mutated cancer cells[67]. Thus, in cancer cells that are dependent on STAT3 for survival, it may be difficult for survival to be maintained by continuous activation of JAK alone. In this context, methylation of STAT3 by PRMT5 can promote activated tyrosine phosphorylation of STAT3, which may work to maintain continuous STAT3 activation in cancer cells. We also found that PRMT5 plays an important role in the oncogenic pathway induced by another STAT3 activation pathway: SRC. Thus, PRMT5 may maintain STAT3 activation in a variety of cancer types. In addition to these findings, PRMT5 expression itself was induced by STAT3. The existence of such a positive feedback loop maintains bistability by supporting the activated state[68]. Thus, this positive activation loop may be important for accelerating and maintaining STAT3 activity in cancer cells. In addition, methylation of SMAD7 by PRMT5 can reportedly promote the binding of SMAD7 to IL-6 receptor glycoprotein 130, resulting in the induction of STAT3 activation. Additionally, PRMT5 inhibition can suppress tumour growth of lung cancer cells in mice[69]. However, as mentioned above, JAK activation alone cannot explain the level of STAT3 activation observed in cancer. It is likely that this mechanism also works in cooperation with direct STAT3 methylation. Furthermore, PRMT5 expression levels have been reported to be upregulated in various cancers. PRMT5 can promote cancer through epigenetic regulation via histone methylation or through activating cancer-related proteins, such as C-MYC and NF-κB, and inhibiting the tumour suppressor p53 by direct methylation[37,70,71]. In this study, we found that STAT3 can induce PRMT5 expression, suggesting that cancer-promoting functions of PRMT5 other than STAT3 activation are involved in the oncogenic function of STAT3. Therefore, further analysis of the regulation of other oncogenic signals and cancer-associated transcription factors via PRMT5 induced by STAT3 should provide insights into the molecular mechanisms underlying the oncogenic functions associated with the PRMT5/STAT3 activation loop.

To date, a variety of STAT3-targeted cancer therapies have been developed[24]. However, there are no clinically applied drugs that inhibit STAT3 itself. Clinical trials of PRMT5 inhibitors are now underway[72], but it is unclear whether they will be safe and effective given the broad physiological effects of PRMT5[37]. Thus, the activation of STAT3 by the PRMT5/MEP50/STAT3 complex discovered in this study may be a new and specific target for suppressing STAT3 for cancer therapeutic approaches.

## Methods
### Cell culture
HEK293T and HeLa cells were purchased from the JCRB Cell Bank. Lenti-X 293 T cells were purchased from Takara Bio (Shiga, Japan). NIH3T3, BEAS2B, A549, and HCC827 cells were purchased from American Tissue Culture Collection (ATCC). *p53*-knockout mouse embryonic fibroblasts ($p53^{-/-}$ MEFs) were obtained as described elsewhere[73]. Mycoplasma contamination was routinely monitored using MycoStrip (InvivoGen, San Diego, CA, USA). HEK293T, HeLa, NIH3T3, and $p53^{-/-}$ MEFs were cultured in Dulbecco's Modified Eagle Medium (DMEM; Nacalai Tesque, Kyoto, Japan) supplemented with 10% fetal bovine serum (FBS; Nichirei Biosciences, Tokyo, Japan). Lenti-X 293 T cells were cultured in DMEM (high glucose) supplemented with 10% FBS. BEAS2B cells were cultured in DMEM/F-12K (Nacalai Tesque) supplemented with 10% FBS. A549 and HCC827 cells were cultured in RPMI-1640 medium (Nacalai Tesque) supplemented with 10% FBS. Streptomycin (100 U/ml)/Penicillin-G (100 µg/ml) solution (Nacalai Tesque) was added to all cell culture media.

### Reagents, antibodies, and plasmid construction
Recombinant IL-6, recombinant soluble IL-6 receptor (sIL-6R), and recombinant HGF were purchased from R&D Systems (Minneapolis, MN, USA). Polybrene was purchased from Nacalai Tesque. Puromycin and hygromycin were purchased from Sigma-Aldrich. Doxycycline was purchased from FUJIFILM Wako Pure Chemical (Osaka, Japan). For stable luciferase expression by recombinant lentivirus, CMV-GFP-T2A-luciferase plasmid was purchased from System Biosciences (Palo Alto, CA, USA). The antibodies used in this study are listed in Supplementary Table 1. For the construction of recombinant lentivirus-mediated shRNA expression plasmid, synthesized oligo was inserted into the pLKO.1 puro vector (Addgene; plasmid #8453). For recombinant lentivirus-mediated inducible shRNA expression, the synthesized oligo was inserted into the Tet-pLKO-puro vector (Addgene; plasmid #21915). Knockdown efficiency was examined using western blot analysis. For STAT3 knockout using the CRISPR/Cas9 system, the synthesized oligo was inserted into the pLenti-CRISPR-

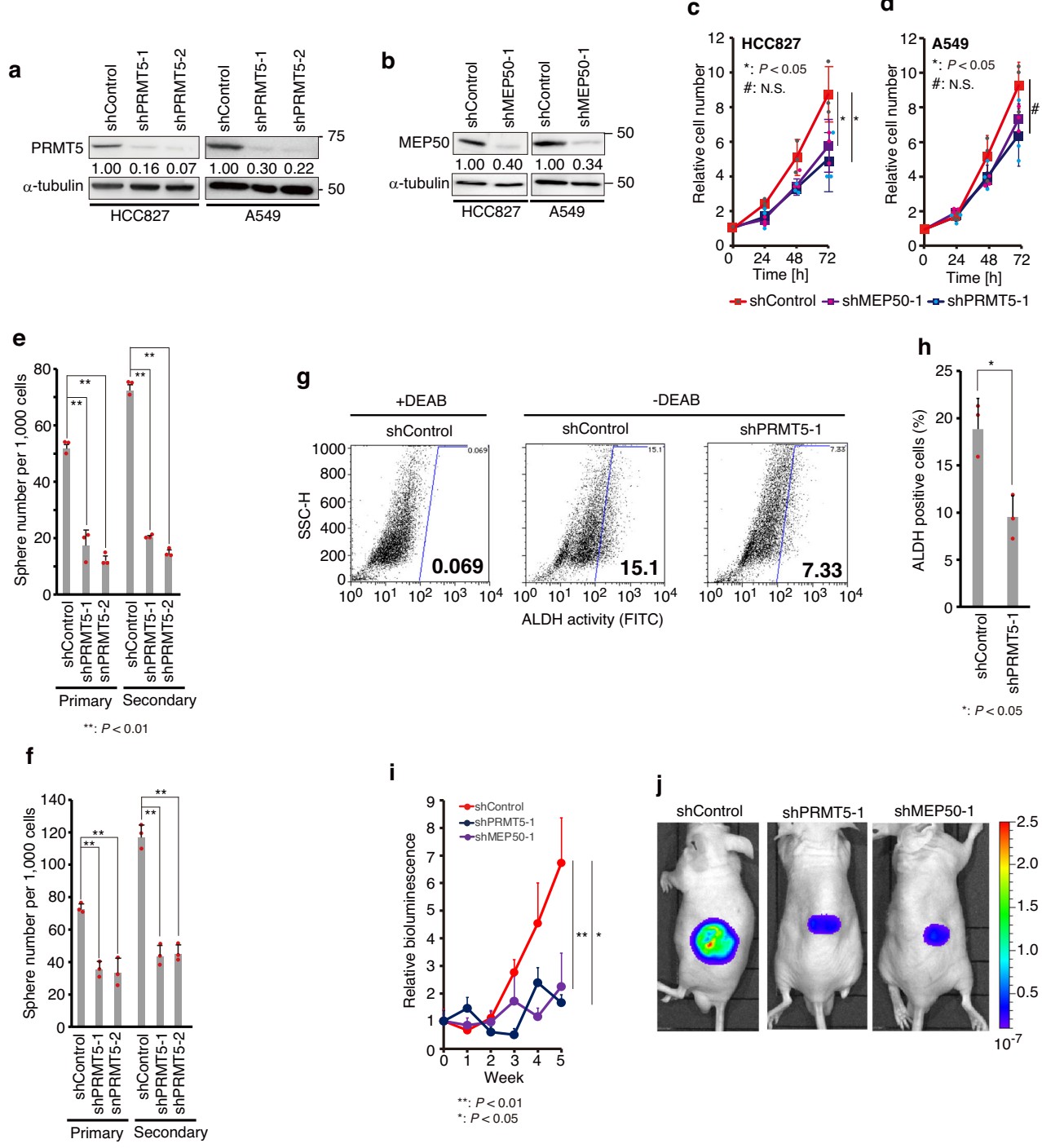

V2-Puro vector (Addgene; #98290). Cas9 and guide RNA for STAT3 were stably expressed by recombinant lentivirus. Knockout efficiency was examined using western blot analysis using an anti-STAT3 antibody. The target sequences for RNAi-mediated knockdown are listed in Supplementary Table 2. The target sequences for CRISPR/Cas9-mediated knockout are listed in Supplementary Table 3. For STAT3 knockout, we prepared two guide RNAs. Myc-tagged STAT3 mutants were generated by PCR, and the relevant primers are listed in Supplementary Table 4. Myc-tagged STAT3 and its mutant cDNAs were inserted into the pcDNA3 vector for transient expression (Thermo Fisher Scientific, Waltham, MA, USA) or pLVSIN-

Hyg vector to produce recombinant lentivirus (Takara Bio) using the In-Fusion HD cloning kit (Takara Bio), in accordance with the manufacturer's instructions.

**Transfection, recombinant virus production, and infection**

The expression plasmid was transfected into cells using GeneJuice (Merck Millipore, Burlington, MA, USA), in accordance with the manufacturer's instructions for transient gene expression. Recombinant retrovirus production was performed as described previously[34]. Recombinant lentivirus was produced in Lenti-X 293 T cells by transfection of the expression

**Fig. 3 | PRMT5 is involved in tumour formation through cancer stem cell (CSC) maintenance. a, b** Expression of **a** PRMT5 and **b** MEP50 was stably knocked down in HCC827 and A549 cells. Short hairpin RNAs (shRNAs) targeting PRMT5 and MEP50 were expressed by recombinant lentivirus. Protein levels were detected by western blotting using the antibody for PRMT5 (abcam; ab10941) or MEP50 (abcam; ab154190). PRMT5 or MEP50 protein levels were quantified with normalization to the β-actin expression intensity. **c, d** Knockdown of **c** PRMT5 and **d** MEP50 attenuated cell proliferation. First, 50,000 cells were seeded on a 12-well plate in triplicate. Viable cells were counted by trypan blue exclusion. Results are shown as the mean ± standard deviation (SD) from three experiments. Tukey's honestly significant difference test was applied for statistical comparisons. The relative cell number indicates the total cell number/Total cell number at time 0 of each cell. **e, f** PRMT5 is involved in CSC maintenance. First, 1000 cells were seeded on a six-well ultra-low-attachment plate in triplicate, and then spheres were counted after 7 days of growth. Results are shown as the mean ± SD from three experiments. **e** Sphere number from HCC827 cells. **f** Sphere number from A549 cells. In **e** and **f**, Tukey's honestly significant difference test was applied for statistical comparisons. **g, h** Knockdown of PRMT5 attenuates the CSC population in HCC827 cells. The population of CSCs was assessed by examining aldehyde dehydrogenase (ALDH)

activity using flow cytometry. **g** Representative flow cytometry plots indicate side scatter (SSC) versus intensity of FITC fluorescence. ALDH activity was based on FITC intensity. Cells having high ALDH activity were plotted in the gated area. A group treated with an ALDH inhibitor, 4-diethylaminobenzaldehyde (DEAB), was used as a negative control. The gating strategy is shown in Supplementary Fig. 3c. **h** The CSC population is shown as the mean ± SD from three independent experiments. Unpaired two-tailed Student's t-test was applied for statistical comparisons. **i, j** Knockdown of PRMT5 attenuates tumour growth. A549 cells with shPRMT5 or shMEP50 expression induced by the Tet-ON system were established. These cells also stably expressed luciferase by recombinant lentivirus. shRNAs for PRMT5 or MEP50 inducible cells were transplanted into 7-week-old nude mice at $1 \times 10^6$ cells ($n = 4$). When tumour volume reached approximately 150 mm³, doxycycline (1 mg/mL) administration was started. Tumour volume was measured by In Vivo Imaging System (IVIS). The individual datapoints are in Supplementary Fig. 3f. Representative IVIS images are shown in **j** (4 weeks). Results are shown as the mean ± standard error of the mean (SEM). Tukey's honestly significant difference test was applied for statistical comparisons. The individual datapoints are in Supplementary Fig. 3f. Source data are provided in Supplementary Data 1. Full immunoblot images are shown in Supplementary Fig. 7.

plasmid and Lentiviral High Titer Packaging Mix (Takara Bio) using TransIT-Lenti transfection reagent (Mirus Bio, Madison, WI, USA), in accordance with the manufacturer's instructions. Twenty-four hours after transfection, the medium was changed, and the cells were incubated for an additional 24 h. The lentivirus-containing medium was collected, after which debris was removed using a syringe filter (pore size 0.45 µm; CORNING, Corning, NY, USA). The filtered culture media were supplemented with 8 µg/mL polybrene to infect cells. Infected cells were selected using puromycin (2.5 µg/mL; Sigma-Aldrich, St. Louis, MO, USA) for shRNA-expressing, v-SRC-expressing cells and STAT3 knockout cells or hygromycin (200 µg/mL; Sigma-Aldrich) for Myc-tagged STAT3-expressing cells.

### Immunoprecipitation, cell fractionation, and immunoblotting

Cells were lysed in TNE lysis buffer [1% Nonidet P-40, 10 mM Tris-HCl (pH 8.0), 150 mM NaCl, 1 mM EDTA, and protease inhibitor cocktail (Nacalai Tesque)]. In accordance with the manufacturer's instructions, cell fractionation was performed using the LysoPure Nuclear and Cytoplasmic Extraction Kit (FUJIFILM Wako Pure Chemical). Protein concentrations were measured using Protein Assay Dye Reagent (Bio-Rad, Hercules, CA, USA). Equal amounts of proteins were used for immunoprecipitation or immunoblotting experiments. Immunoprecipitation was performed as described previously[35]. Cell lysates were incubated with the indicated antibody for 2 h at 4 °C. Protein-G Sepharose beads (Cytiva, Marlborough, MA USA) were added to the cell lysate, which were then incubated for 2 h at 4 °C. Next, the protein-G beads were collected and washed with lysis buffer before denaturation. SDS-PAGE was performed with sample loading at 30 µg per well, followed by immunoblotting. The proteins separated on the SDS-PAGE gel were transferred to a polyvinylidene fluoride (PVDF) membrane (Merck-Millipore), followed by blocking with 3.5% bovine serum albumin (BSA; Nacalai Tesque) in PBS with 0.05% Tween-20 (FUJIFILM Wako Pure Chemical). The blots were then incubated with the primary antibody for 40 min at room temperature or 4 °C overnight, and the horseradish peroxidase (HRP)-conjugated secondary antibody for 1 h at room temperature (Cytiva). Primary and secondary antibodies were diluted with Bullet Immuno-Reaction Buffer (Nacalai Tesque) or 3.5% BSA (blocking solution). Information on the antibodies is presented in Supplementary Table 1. Chemi-Lumi One Ultra (Nacalai Tesque) and LuminoGraph I (ATTO, Tokyo, Japan) were used to visualize the blots. Quantification of protein expression levels was performed using ImageJ software.

### Cell proliferation assay, tumour sphere formation, and CSC population

First, $5 \times 10^4$ cells were seeded per well in a 12-well plate in triplicate. Twenty-four hours later, viable cells were measured by Trypan Blue exclusion using a

cell counter (Bio-Rad). For tumour sphere formation, 1,000 cells were seeded per well in a six-well ultra-low-attachment plate (CORNING). The cells were maintained in DMEM/F12K supplemented with 20 ng/mL human recombinant epidermal growth factor (EGF; R&D Systems, Minneapolis, MN, USA) and 20 ng/mL human recombinant basic fibroblast growth factor (bFGF; FUJIFILM Wako Pure Chemical). Seven days after seeding, the tumour spheres with a diameter >150 µm were counted. The population of CSCs was assessed using a FACSCalibur (BD, Franklin Lakes, NJ, USA) flow cytometer with an ALDEFLUOR kit (Stem Cell Technology, Vancouver, Canada).

### RNA isolation, reverse transcription, and quantitative PCR (qPCR)

Total RNA isolation was performed using NucleoSpin RNA Plus (Takara Bio), after which RNA concentrations were measured using a NanoDrop (Thermo Fisher Scientific, Waltham, MA, USA). A total of 500 ng of RNA was used for reverse transcription with the PrimeScript RT reagent kit (Takara Bio). cDNA templates were subjected to qPCR analysis using the TaqMan Gene Expression Assay Mastermix and TaqMan probes (Thermo Fisher Scientific) with a StepOnePlus Real-Time PCR System (Thermo Fisher Scientific). The probe list for the TaqMan gene expression assays is shown in Supplementary Table 5. Each amplification reaction was performed in triplicate, and the average of three threshold cycles was used to calculate the relative amount of transcript in the sample (StepOne Software; Thermo Fisher Scientific). mRNA was quantified and expressed in arbitrary units as the ratio of the sample quantity to the calibrator or the mean values of control samples. All values were normalized to an endogenous control, the human β-actin gene.

### Immunofluorescence staining

First, $5 \times 10^4$ cells were seeded per well in a 24-well glass-bottomed plate (AGC TECHNO GLASS, Shizuoka, Japan). Cells were fixed with 4% paraformaldehyde for 15 min at 37 °C and permeabilized with 0.5% TritonX-100 in PBS for 7 min at room temperature. Blocking was performed using 1% BSA, followed by primary antibody incubation (anti-PRMT5 antibody or anti-STAT3 antibody) for 30 min at room temperature. Secondary antibodies [Alexa-488-conjugated anti-mouse antibody (Thermo Fisher Scientific) and Alexa-568-conjugated antibody (Thermo Fisher Scientific)] were incubated for 30 min at room temperature in the dark. DNA was stained with Hoechst33342 (Thermo Fisher Scientific). Immunofluorescent images were recorded using a FLUOVIEW FV1200 biological laser scanning microscope (EVIDENT, Tokyo, Japan). Images were captured using FLUOVIEW software (ver. 3.1.1; EVIDENT).

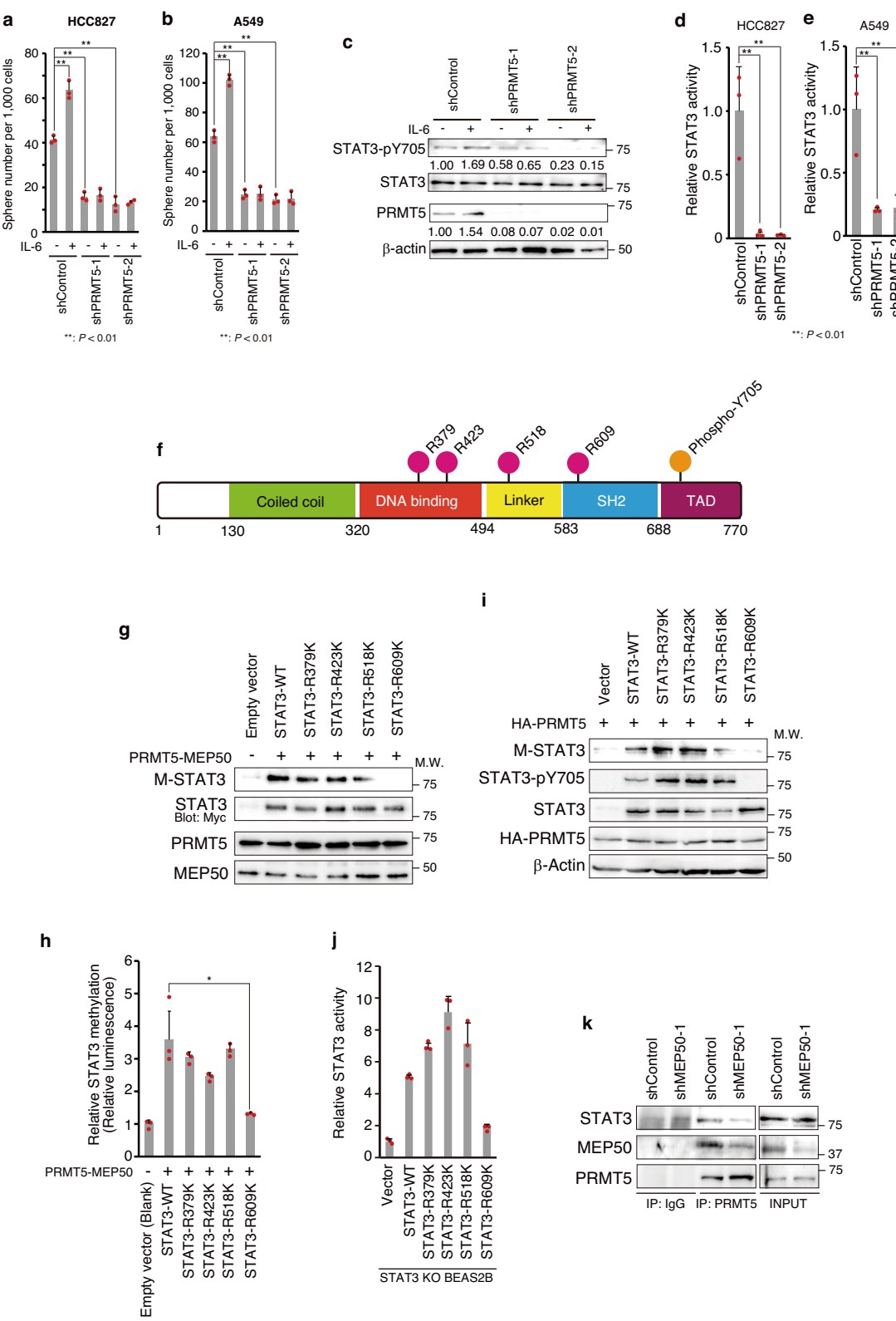

## Luciferase assay

Luciferase assays were performed as described previously[74]. In accordance with the manufacturer's instructions, Cignal Reporter Assay Kit STAT3 (QIAGEN, Hilden, Germany) was used as a STAT3 reporter plasmid. The HIF-1 reporter plasmid [pGL4.42 (luc2P/HRE/ Hygro) vector] was purchased from Promega (Madison, WI, USA), and the NF-κB reporter plasmid used in this study was described[73]. Forty-eight hours after transfection, reporter activity was assessed by GLOWMAX (Promega) with the Dual-luciferase reporter assay system (Promega).

**Fig. 4 | PRMT5 upregulates STAT3 activity by PRMT5-mediated methylation.**
**a, b** Sphere numbers with IL-6 treatment in shControl- and shPRMT5-expressing
**a** HCC827 and **b** A549 cells. Cells were treated with recombinant IL-6 (50 ng/mL)
and recombinant sIL-6 (50 ng/mL) for 7 days, after which the numbers of spheres
were determined. The results are shown as the mean ± standard deviation (SD) from
three experiments. Tukey's honestly significant difference test was applied for sta-
tistical comparisons. **c** Western blot analysis of STAT3 activation after 50 ng/ml
recombinant IL-6 and 50 ng/ml recombinant sIL-6 treatment for 7 days in control
and PRMT5-knockdown cell-derived spheres. PRMT5 and STAT3 protein levels
were detected by western blotting using an antibody for PRMT5 (abcam; ab10941)
or STAT3 (CST; 12640). STAT3 activation was monitored by western blotting using
an anti-phosphorylated STAT3 (Y705) antibody (CST; 9138). PRMT5 expression
levels were quantified with normalization to β-actin expression intensity and
phosphorylated STAT3 intensity was normalized to STAT3 expression intensity. **d,
e** STAT3 transcriptional activity in PRMT5-knockdown **d)** HCC827 and **e)** A549
cells. STAT3 activity was measured by luciferase assays using a STAT3 reporter
plasmid. Results are expressed as firefly/*Renilla* ratios and presented as the
mean ± SD from three experiments. Tukey's honestly significant difference test was
applied for statistical comparisons. **f** Schematic diagram of arginine residues
methylated by PRMT5 in human STAT3. TAD: Transcriptional Activation Domain.
**g** In vitro methylation of STAT3. Myc-tagged wild-type STAT3 (STAT3 WT) or its
mutants were transiently expressed in HEK293T cells and purified by immuno-
precipitation using an antibody for Myc-tag. Methylation of STAT3 by recombinant
human Flag-tagged PRMT5/His-tagged MEP50 complex was detected by western
blotting using an anti-symmetric dimethyl arginine antibody (Merck Millipore; 07-
413). STAT3, PRMT5, and MEP50 protein levels were determined by western

blotting using an antibody for the indicated tag [STAT3 (Myc-tag): GeneTex
(GTX115046), PRMT5 (Flag-tag): SIGMA (SAB4301135), MEP50 (His-tag): Pro-
teintech (10001-0-AP)]. **h** Quantification of PRMT5/MEP50 complex-mediated
STAT3 methylation. Methylated STAT3 was quantified using MTase-Glo™
Methyltransferase assay kit (Promega). The results are shown as the mean ±
standard deviation (SD) from three experiments. Unpaired two-tailed Student's
t-test was applied for statistical comparisons. **i** Methylation of arginine residue 609
(R609) in STAT3 by PRMT5 in STAT3-KO BEAS2B cells. STAT3 was knocked out
using the CRISPR–Cas9 system. Myc-tagged STAT3 wild type (WT) and its mutants
were transiently expressed in STAT3-knockout BEAS2B cells. Methylated STAT3
was detected using an anti-symmetric dimethyl arginine antibody (Merck Millipore;
07-413). Expression of HA-tagged PRMT5 was detected using an anti-HA antibody
(abcam; ab130275), and Myc-tagged STAT3 was detected using a STAT3 antibody
(CST; 9139). **j** Methylation of STAT3 R609 is necessary for its activation. STAT3
activity was measured by luciferase assays. STAT3 reporter plasmid and Myc-tagged
STAT3 WT and its mutants were transiently expressed in STAT3-KO BEAS2B cells.
Results are expressed as firefly/*Renilla* ratios and presented as the mean ± standard
deviation (SD) from three experiments. **k** Interaction of PRMT5 and STAT3 via
MEP50. The PRMT5–MEP50–STAT3 complex was detected by immunoprecipi-
tation using an anti-PRMT5 antibody (abcam; ab10941). PRMT5-bound MEP50 or
STAT3 was detected by western blotting using an anti-MEP50 antibody (Abcam;
ab154190) or an anti-STAT3 antibody (CST; 9139). Representative results from two
independent experiments are shown in a–e and g–j, while those from three inde-
pendent experiments are shown in k. Source data are provided in Supplementary
Data 1. Full immunoblot images are shown in Supplementary Fig. 7.

## Chromatin immunoprecipitation (ChIP)-qPCR

The ChIP assay was performed using the ChIP-IT Express Enzymatic kit
(Active Motif, Carlsbad, CA, USA) in accordance with the manufacturer's
instructions. Immunoprecipitation was recovered with Protein-G-
conjugated magnetic beads (included in the ChIP-IT Express Enzymatic
kit) together with STAT3 antibody (12640, 1:50; CST, Danvers, MA USA)
or Rabbit IgG XP Isotype control (3900, 1:50; CST) at 4 °C for 16 h with
gentle rotation. The immunoprecipitated DNA was purified using the
Chromatin IP DNA Purification Kit (Active Motif). Purified DNA was
subjected to qPCR analysis using SYBR Green Real-Time PCR Master Mix
-Plus- (Toyobo, Osaka, Japan). Oligonucleotides used for ChIP-qPCR are
described in Supplementary Table 6.

## In vitro methylation assay

Plasmids expressing the Myc-tagged wild-type STAT3 and its mutants were
transfected into HEK293T cells. Transiently expressed STAT3 was purified
by immunoprecipitation using an anti-Myc tag antibody and Protein G
Sepharose beads (Cytiva). The immunoprecipitated STAT3 was washed
twice with a reaction buffer [20 mM Tris-HCl (pH 8.0), 50 mM NaCl, 3 mM
$MgCl_2$, 1 mM DTT, 1 mM EDTA, and 0.1 mg/ml BSA]. Then, the STAT3-
bound Protein G beads were suspended in 50 μl of reaction buffer. One
hundred nanograms of recombinant active Flag-tagged PRMT5/His-tagged
MEP50 complex (Sigma-Aldrich; SRP0145), 100 mM symmetric adenosyl
methionine (BPS Bioscience, San Diego, CA, USA; 52120), and 10 μl of
STAT3-bound Protein G bead suspension were added to a 50 μl reaction
volume. The reaction mixture was incubated for 1 h at 30 °C. The enzymatic
reaction was stopped by adding 4× SDS-PAGE sample buffer (Bio-Rad).
Methylated STAT3 was detected by western blotting using a symmetric
dimethyl arginine antibody.

 STAT3 methylation was also detected and quantified using the MTase-
Glo™ Methyltransferase assay kit (Promega), in accordance with the
manufacturer's instructions. The PRMT5/MEP50 complex and STAT3
reaction was performed under the same conditions as described above.

## Mouse xenograft model

The animal experiment committee at Nippon Medical School approved the
animal experiments in this study (Approval Number: 29-045, 2022-044),
and animal care was conducted following the Guidelines for Animal

Experiments of Nippon Medical School and the guidelines of The Law and
Notification of the Government of Japan[75], as well as the ARRIVE guide-
lines. Mice were maintained under a 12-h light/12-h dark cycle at 20–24 °C
with 40%–70% humidity. They were given free access to standard laboratory
mouse chow (MF; Oriental Yeast,Tokyo, Japan) and drinking water. They
were housed with a maximum of five mice per cage. All mice were checked
for stress each day. Nude mice (BALB/c-AJcl nu/nu) were purchased from
Japan CLEA, Inc. (Tokyo, Japan). Luciferase was stably expressed by
recombinant lentivirus in cells before transplantation. Then, $1×10^6$
(Figs. 3i, j) or $2×10^6$ (Fig. 5f, g) cells were subcutaneously transplanted into 6
or 7-week-old male nude mice (body weight 23–26 g). Tumour volume was
assessed using In Vivo Imaging System (IVIS; Revvity, Waltham, MA USA).
For the induction of shRNA expression in xenografted tumours, tumour
volume was allowed to reach approximately 150 $mm^3$, after which dox-
ycycline (1 mg/mL) was added to the drinking water.

## Bioinformatic analysis

The differential expression of *PRMT5* between tumour sites and normal
sites was analysed using the TCGA-LUAD dataset (https://portal.gdc.
cancer.gov/projects/TCGA-LUAD) or the TCGA-LUSC dataset (https://
portal.gdc.cancer.gov/projects/TCGA-LUSC). Overall survival rates were
compared between the groups with high and low *PRMT5* expression and the
correlation of *PRMT5* copy number with *PRMT5* expression in LUAD was
also examined using the TCGA-LUAD dataset. Datasets were analysed
using the UCSC Xena browser (https://xena.ucsc.edu). *PRMT5* expression
levels across 13 cancer types were examined using TCGA PanCancer Atlas
Studies. The dataset was analysed on cBioPortal (http://www.cbioportal.
org). Differential *PRMT5* expression in cancer cell lines was examined using
the Depmap portal in Cancer Cell Line Encyclopedia (CCLE; https://sites.
broadinstitute.org/ccle) at the Broad Institute. The ChIP-seq dataset was
obtained from Gene Expression Omnibus (GEO) at NIH National Center
for Biotechnology Information (NCBI) (https://www.ncbi.nlm.nih.gov/
geo/) and was analysed using the UCSC Genome Browser (https://genome.
ucsc.edu). The PRMT5-mediated methylation of STAT3 arginine residues
was predicted using PRmePRed (http://bioinfo.icgeb.res.in/PRmePRed/
index.html). Sites with a prediction score of > 0.8 were considered to be
candidates for PRMT5-mediated methylation of arginine residues
in STAT3.

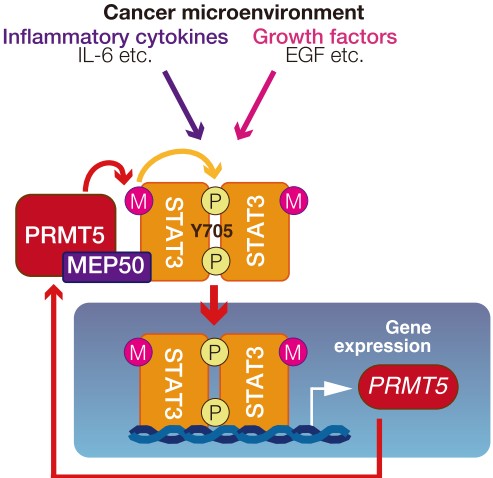

## Statistical analysis and reproducibility

Statistical analysis was conducted using StatFlex Ver. 6.0 (Artech, Osaka, Japan) or Microsoft Excel. In vivo analysis results are presented as the mean ± standard error of the mean (SEM), whereas in vitro analysis results are expressed as the mean ± standard deviation (SD). Tukey's honestly significant difference test was applied for statistical comparisons (excluding results in Figs. 2f, 2h, 3h, 5c, and Supplementary Fig. 5f), or unpaired two-tailed Student's t-test was used. A P-value of < 0.05 was considered statistically significant. Details regarding N numbers and data reproducibility can be found in the figure legends.

**Fig. 5 | PRMT5-mediated STAT3 methylation of arginine residue 609 is necessary for STAT3 activation and tumour formation through cancer stem cell (CSC) maintenance. a** Phosphorylation at STAT3 tyrosine residue 705 is inhibited in the STAT3 mutant (R609K). Myc-tagged STAT3 WT and STAT3 R609K mutants were stably expressed by recombinant lentivirus in STAT3-knockout HCC827 cells. Endogenous STAT3 and Myc-tagged STAT3 were detected by western blotting using a STAT3 antibody (CST; 9139) or Myc-tag antibody (abcam; ab32). STAT3 activation was monitored by western blotting using an anti-phosphorylated STAT3 (Y705) antibody (Abcam; ab267373). Other protein expression levels were also determined by western blotting using antibodies against the indicated antibodies. The lower band of STAT3 was seen in Myc-STAT3-expressing cells. This band was not seen with the Myc-tag antibody, suggesting that the N-terminal part containing the tag was specifically degraded for Myc-STAT3. **b** After IL-6-mediated STAT3 activation, the STAT3 R609K mutant failed to translocate to the nucleus. 50 ng/ml recombinant IL-6 and 50 ng/ml recombinant sIL-6 were treated for 8 h. Cells were lysed and separated into cytosolic and nuclear fractions. Nuclear or cytosolic STAT3 was detected by immunoblotting using antibodies against STAT3 (CST; 9139). STAT3 activation was monitored by western blotting using an anti-phosphorylated STAT3 (Y705) antibody (CST; 9138). Lamin A/C, a nuclear marker, was detected by western blotting using an anti-Lamin A/C antibody (Santa Cruz; sc-7293), and α-tubulin, a cytoplasmic marker, was detected by western blotting using an α-tubulin

antibody (Sigma-Aldrich; T6199). **c** ChIP-qPCR analysis revealed that the STAT3 R609K mutant did not directly bind to the regulatory region of *PRMT5* in HCC827 cells. Results are presented as the mean ± SD from three experiments. Unpaired two-tailed Student's t-test was applied for statistical comparisons. **d, e** Methylation of R609 is necessary for **d)** cancer cell proliferation and **e)** sphere formation. In **d**, 50,000 cells were seeded in triplicate on a 12-well plate. Viable cells were counted by trypan blue exclusion. In **e**, 1,000 cells were seeded in triplicate on a six-well ultra-low-attachment plate, and then spheres were counted after 7 days of growth. Results are shown as the mean ± SD. **f, g** STAT3-R609K mutant expression failed to recover tumour formation ability. Myc-tagged STAT3 WT, STAT3 Myc-tagged STAT3 R609K, and luciferase were stably expressed by recombinant lentivirus in STAT3-knockdown HCC827 cells. Then, 2×10⁶ cells were transplanted into 6-week-old male nude mice (n = 5). Tumour volume was measured by IVIS. Results are shown as the mean ± standard error of the mean (SEM). Representative IVIS images (4 weeks) are shown in **g**. The individual datapoints are in Supplementary Fig. 6. In **d–f**, Tukey's honestly significant difference test was applied for statistical comparisons. **h** Schematic illustration of the STAT3–PRMT5 circuit in lung adenocarcinoma cells. Representative results from two independent experiments are shown in d and e. Source data are provided in Supplementary Data 1. Full immunoblot images are shown in Supplementary Fig. 7.

## Data availability

All materials used in this study are available with permission from the corresponding author. Data for PRMT5 or MEP50 expression, PRMT5 copy number and overall survival (OS) in lung adenocarcinoma was obtained from the cohort of GDC TCGA Lung Adenocarcinoma (15 datasets: https://xenabrowser.net/datapages/?cohort=GDC%20TCGA%20Lung%20Adenocarcinoma%20(LUAD)&removeHub=https%3A%2F%2Fxena.treehouse.gi.ucsc.edu%3A443). PRMT5 expression or lung squamous cell carcinoma was obtained from the cohort of GDC TCGA Lung Squamous Cell Carcinoma (15 datasets: https://xenabrowser.net/datapages/?cohort=GDC%20TCGA%20Lung%20Squamous%20Cell%20Carcinoma%20(LUSC)&removeHub=https%3A%2F%2Fxena.treehouse.gi.ucsc.edu%3A443). In Fig. 1c, protein levels of PRMT5 and STAT3 were obtained from Proteomics data of PRMT5 (O14744) and STAT3 (P40763) at the depmap portal. In Supplementary Fig. 1g, *PRMT5* expression level was obtained from expression public 23Q4 at the depmap portal. In Supplementary Fig. 1g, *PRMT5* expression data across cell lines was obtained from CCLE RNAseq gene expression data for 1019 cell lines at Cancer Cell Line Encyclopedia (CCLE). In Supplementary Fig. 2d, ChIP-seq data of STAT3 or H3K27Ac in H358 cells was obtained from GEO at NCBI (STAT3 ChIP-seq data: GSM2752894, H3K27Ac ChIP-seq data: GSM1635574). Original immunoblot images were shown in Supplementary Fig. 7 (Main Figs. 2–5) and 8 (Supplementary Figs. 2, 4 and 5). The source data behind the graph is shown in Supplementary Data 1 (Main Figures) and 2 (Supplementary Figures.).

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

## Acknowledgements
We thank Wataru Nakajima and Ikuno Uehara for the helpful discussions and for the technical assistance from Akihiro Yamazaki, Yumi Asano, and Mitsuko Kajita. We thank T. Akagi for providing us with the pBabe-v-Src expression plasmid. We also thank J. Iacona, Ph.D., from Edanz (https://jp.edanz.com/ac) for editing a draft of this manuscript. This work was supported by JSPS KAKENHI (Grant Numbers 25870792, 20K09044 and 23K06641) and The Kurata Grants by The Hitachi Global Foundation.

## Author contributions
T.S., Y.A., and N.T. conceived and designed the experiments. T.S. and Y.A. conducted the experiments with assistance from N.O. and M.O. Finally, N.T. and Y.A. wrote the manuscript. All authors contributed to reviewing the manuscript.

## Competing interests
The authors declare no competing interests.
