## [Peer Review File · Communications Biology]

Reviewers' comments:

Reviewer #1 (Remarks to the Author):

This manuscript focuses on post-translational modification of STAT3 by PRMT5. The initial figures shows that PRMT5 is over expressed in primary NSCLCs and cell lines, and high expression correlates with poor prognosis in lung adenocarcinomas. The authors then knock-down PRMT5 or STAT3 and focus on protein expression, cell growth in 2D and 3D and cell growth in vivo. The authors identified arginine 609 as the methylated residue and perform a series of experiments in one NSCLC cell line to demonstrate that mutation of this residue prevents rescue of 3D and in vivo cell growth in STAT3 KO cells.

While the modification of STAT3 by PRMT5 may be of importance, the cell line data are not fully convincing of the pathways proposed. There are also several instances of missing information or controls that would need to be addressed.

Point 1: It is unclear if all the NSCLC cells line in CCLE are indicated in figure 1, or only selected lines. There is no correlation analysis between STAT3 and PRMT5 expression in either primary tumors or cells lines. PRMT5 expression has already been shown to be high in NSCLC.

Point 2: The ENCODE and PCR data show that STAT3 can bind the PRMT5 promoter, but the ChIP-PCR should be performed with real time PCR and quantified in a lung cancer cell line rather than in NIH3T3 cells.

Point 3: It is unclear if PRMT5 expression is truly changed by STAT3 knock down due to uneven loading of the western blot in Figure 2. The reduction of PRMT5 expression after IL6 or EGF challenge by STAT3 knock-down should be done in a lung cancer line instead of NIH3T3 cells.

Point 4: In figure 4, the stability of STAT3 protein appears to be the mechanism, rather than STAT3 phosphorylation. The pSTAT3 should be normalized to total STAT3 protein levels.

Point 5: The STAT3 "KO" cells are used in the final figure, and form surprisingly no 3D spheroids and very little tumor. However, in the text they are referred to as knock-down, and there is no WB for this line or detailed description of the selection and verification of KO clones. There is no demonstration that the rescue of STAT3 was equivalent to parental line levels for both the WT and mutant.

Point 6: Demonstration of KD of proteins by Dox inducible shRNA constructs is lacking. There is no assessment of tumor histology and therefore the data lack depth.

Point 7: Cancer stem cell properties may be influenced, but growth is also dramatically changed. The authors could soften language to include both stemness and proliferation as potential phenotypes.

Minor points

1. The beginning of the results reads as more introduction.
2. The MEFs with p53 KO are listed as 'described elsewhere' with no reference in the methods.

3. The graphs of cell growth (3C, 3D) do not indicate the y-axis value in detail - relative cell number is percent of initial?
4. There are no statistics on graph 3K.
5. The micrographs in Fig. 3F and 3H appear very different, while the data on the graphs are similar.
6. The pLKO.1 vector encodes shRNAs with a stem-loop structures rather than siRNA oligos.
7. SOX2 is a lineage transcription factor and may indicate a switch towards more squamous fate rather than stemness. Given SOX2 and OCT4 were not assayed, their mention appears tangential.
8. Line 373 should read 'Trypan Blue exclusion'

Reviewer #2 (Remarks to the Author):

Sano and colleagues investigated the interplay between PRMT5 and STAT3. By using a series of cellular and biochemical assays they conclude that STAT3 regulates PRMT5 expression. Additionally, PRMT5 methylates STAT3, a post-transcriptional modification that is needed for full activation of STAT3 and for its tumorigenic potential.

Major points

- 1) Line 101: EGFR is not the most often mutated oncogene in NSCLC.
- 2) Across the whole manuscript and figures, it is not clear if the Authors are using siRNA or shRNAs. From the methods, it looks that they are using shRNAs expressed from lentivirus, but across the whole paper they use the expression siXXXX. Also sometimes, they refer to a “knock-down” when using the Crispr/Cas9 system (most likely they mean “knockout”). The authors mix the terms knockout and knockdown across the figures and text. This makes it difficult to interpret the results.
- 3) Across the manuscript different experiments are done in different cell lines. There is not consistency which makes it difficult to interpret the general picture.
- 4) Figure 2: Key findings should be confirmed using a STAT3 knockout cancer cell line. Specifically, the regulation of PRMT5, at mRNA and protein levels, by STAT3 upon stimulation with IL6 and EGF should be confirmed using STAT3 knockout cells.
- 5) Figure 4C: It is difficult to interpret. Knockdown of PRMT5 results in downregulation of total STAT3 levels, which makes it difficult to conclude that there is less STAT3 activation (pY705). In addition, in the control cell line upon stimulation of IL6, there is more total STAT3. Thus, it is difficult to know if the increase of pY705 is due to IL6 or just to more total STAT3. A better WB has to be shown here.
- 6) Figure 4f. M-STAT3 does not detect methylated STAT3, but just a methylated protein that is immunoprecipitated with PRMT5 and has around 80 kDa.
- 7) In general, the direct evidence that PRMT5 methylates STAT3 is quite weak. Figure 5c, as above, M-STAT3 does not detect methylated STAT3, but just a methylated protein of around 80 kDa (indeed, M-

STAT3 seems to be present in the STAT3-KO cells). Previous reports have shown that while PRMT5 may interact with STAT3, in vitro PRMT5 does not methylates STAT3 (Tee et al. Ref 63). Methylation levels of endogenous STAT3 in cell lines using mass spectrometry in presence and absence of PRMT5 should be shown. This is a key experiment on this the paper.

8) Figure 5c STAT3 and pY705 are detected in STAT3 knock out cells.

9) Figure 5e (showing STAT3 levels) is confusing. Parental cell line expressed only STAT3alpha?, "Vector" (STAT3 knock out) still expressed STAT3alpha?, wt and mutant reconstituted cell lines seems to express STAT3 alpha and beta. Or, is the lower band the myc-tagged version? Which usually it should run higher than non myc tagged STAT3? Nevertheless, STAT3 KO cells should not express STAT3. Or it is a knock-down.? In the figure legend is written both, Knockout and Knockdown. M-STAT3 does not really detects methylated STAT3.

10) Assuming the cell lines described in figure 5e are used in the xenograft experiments, it is somehow surprising to see such big differences in tumor growth between the parental and the STAT3 KO (or knock-down?) cells, while the levels of STAT3 in both cell lines are not so different.

11) The co-localization experiments between PRMT5 and STAT3 are difficult to interpret. High resolution, high magnification pictures done with a confocal microscope should be shown. This should be sustained with a Duolink assay or similar.

12) The STAT3 mutant R609K seems to be less phosphorylated (Y705), is it still able to translocate to the nucleus and bind DNA upon IL6 stimulation (compared to wt STAT3)?

13) Across the manuscript the authors compare in several experiments more than two groups. A t-test is not appropriated

Minor points

1) Figure 3m, n seems not to be in the manuscript, or refers to figure 3K i.

2) Sometimes, the figures legends lack of information, such incubation times in the treatments

Responses to the reviewers' comments

MS: COMMSBIO-23-2230

Title: PRMT5-mediated methylation of STAT3 is required for lung cancer stem cell maintenance and tumour growth

Authors: Yoshinori Abe, Takumi Sano, Naoki Otsuka, Masashi Ogawa, and Nobuyuki Tanaka

We are grateful to the referees for their invaluable comments and suggestions. In accordance with their suggestions, we performed additional experiments so as to adequately address the issues raised and have amended the paper accordingly. Please find below our point-by-point responses to each of the comments.

Reviewer #1 (Remarks to the Author):

This manuscript focuses on post-translational modification of STAT3 by PRMT5. The initial figures shows that PRMT5 is over expressed in primary NSCLCs and cell lines, and high expression correlates with poor prognosis in lung adenocarcinomas. The authors then knock-down PRMT5 or STAT3 and focus on protein expression, cell growth in 2D and 3D and cell growth in vivo. The authors identified arginine 609 as the methylated residue and perform a series of experiments in one NSCLC cell line to demonstrate that mutation of this residue prevents rescue of 3D and in vivo cell growth in STAT3 KO cells.

While the modification of STAT3 by PRMT5 may be of importance, the cell line data are not fully convincing of the pathways proposed. There are also several instances of missing information or controls that would need to be addressed.

Point 1: It is unclear if all the NSCLC cells line in CCLE are indicated in figure 1, or only selected lines.

In Fig. 1c (now Supplementary Fig. 1f), we analyzed all CCLE database cells (1000 types). Therefore, we have added the following description into the text: "The ranking of *PRMT5*

expression in all 1,000 cancer cell lines using the Cancer Cell Line Encyclopedia (CCLE; <https://sites.broadinstitute.org/ccle/>) also showed high *PRMT5* expression levels in NSCLC cells (Supplementary Fig. 1f)” (lines 134 to 137).

There is no correlation analysis between STAT3 and PRMT5 expression in either primary tumors or cells lines. PRMT5 expression has already been shown to be high in NSCLC.

According to this comment, we have performed correlation analysis between STAT3 and PRMT5 expression in cells lines obtained from the CCLE. As shown in Fig. 1c and Supplementary Fig. 1g, we found that PRMT5 protein and mRNA expression levels correlated with STAT3 protein expression levels in 12 and 8 NSCLC cell lines, respectively. Therefore, we have added the following description into the text: “Furthermore, using the CCLE, we also found that PRMT5 protein (Fig. 1c) and mRNA (Supplementary Fig. 1g) expression levels correlated with STAT3 protein expression levels in 12 and 8 NSCLC cell lines, respectively, suggesting that STAT3 is involved in PRMT5 expression” (lines 139 to 142).

-

Point 2: The ENCODE and PCR data show that STAT3 can bind the PRMT5 promoter, but the ChIP-PCR should be performed with real time PCR and quantified in a lung cancer cell line rather than in NIH3T3 cells.

In agreement with this comment, ChIP-Seq analysis was performed on NSCLC line H358 in the UCSC Genome Browser. The results are shown in Supplementary Figure 2d and the following description was added to the legend of Supplementary Figure 2d: “**d** The UCSC Genome Browser (<http://genome.ucsc.edu/index.html>) results show the locations of STAT3 ChIP-Seq and acetylated histone H3 at Lys27 (H3K27Ac) ChIP-Seq signals on the PRMT5 locus in H358, which harbors the KRAS mutation (KRAS G12C). Source data are provided in Supplementary Data 3. Full immunoblot images are shown in Supplementary Data 4”.

Moreover, we have added the following description into the text:” **Binding of STAT3 to the PRMT5 gene locus in NSCLC cells was also confirmed by ChIP-Seq analysis in the UCSC Genome Browser database (Supplementary Fig. 2d)”** (lines 165 to 167).

Point 3: It is unclear if PRMT5 expression is truly changed by STAT3 knock down due to

uneven loading of the western blot in Figure 2. The reduction of PRMT5 expression after IL6 or EGF challenge by STAT3 knock-down should be done in a lung cancer line instead of NIH3T3 cells.

According to this comment, we analyzed the induction of PRMT5 expression by STAT3 in NSCLC cells using STAT3 KO A549 and HCC827 cells. As shown in Fig. 2a and b, the induction of PRMT5 expression by IL-6 was markedly suppressed by STAT3 KO. Furthermore, we consider that Western blot loading to be even. From these results we have added the following description into the text: “To further analyse the induction of PRMT5 expression by STAT3 in NSCLC cells, we generated STAT3-knockout (STAT3-KO) cells by using the CRISPR/Cas9 system in A549 (EGFR wild type, adenocarcinoma) and HCC827 cells (EGFR mutant, adenocarcinoma). As expected, the induction of PRMT5 mRNA expression by IL-6 was markedly suppressed by STAT3 KO in A549 and HCC827 cells (Fig. 2a). Moreover, PRMT5 protein expression and induction by IL-6 were also suppressed by STAT3 KO (Fig. 2b), suggesting that PRMT5 is induced by STAT3 in NSCLC” (lines 156 to 162).

Point 4: In figure 4, the stability of STAT3 protein appears to be the mechanism, rather than STAT3 phosphorylation. The pSTAT3 should be normalized to total STAT3 protein levels.

To make this result more definitive, we retried this experiment with different STAT3 antibody. As shown in Fig. 4c, the increase in the amount of STAT3-pY705 normalized to total STAT3 protein level by IL-6 was suppressed by shPRMT5. We believe that our results suggest that tyrosine phosphorylation of STAT3 is upregulated by PRMT5.

Point 5: The STAT3 "KO" cells are used in the final figure, and form surprisingly no 3D spheroids and very little tumor. However, in the text they are referred to as knock-down.

As pointed by the reviewer, this was our writing error, and we have rewritten as follows: “Moreover, STAT3-KO cells had a markedly decreased number of sphere-forming cells, and the R609K mutant showed reduced recovery of sphere-forming cells compared with wild-type STAT3 (Fig. 5e)” (lines 240 to 243).

and there is no WB for this line or detailed description of the selection and verification of KO clones.

In the revised manuscript, we have shown immunoblots of STAT3 in STAT3 KO A549 and HCC827 cells in Fig. 2b and added the selection and verification of KO clones into the Methods (lines 351 to 359, 370 to 374).

There is no demonstration that the rescue of STAT3 was equivalent to parental line levels for both the WT and mutant.

We retried this experiment because the bands were not clear in our original Fig. 5e. As a result, our revised figure (now in Fig. 5a) more clearly shows that the rescue of STAT3 was equivalent to parental line levels for both the WT and mutant.

Point 6: Demonstration of KD of proteins by Dox inducible shRNA constructs is lacking. There is no assessment of tumor histology and therefore the data lack depth.

In accordance with this comment, we have added results showing knockdown of the PRMT5 protein by the Dox-inducible shRNA construct in Supplementary Fig. 3d and e. Because tumor growth was clearly inhibited by Tet-induced reduction of the PRMT5 protein, and tumors in control cells are not inhibited by Tet, we believe that the growth of A549 cell tumors transplanted in nude mice is inhibited by reduction of PRMT5.

Point 7: Cancer stem cell properties may be influenced, but growth is also dramatically changed. The authors could soften language to include both stemness and proliferation as potential phenotypes.

In accordance with this comment, we added the following description into the Discussion: “Furthermore, because STAT3 KO also reduces cell proliferation rates, these results suggest that the induction of STAT3 by PRMT5 is involved in tumorigenesis by regulating both stemness and proliferation” (lines 270 to 272).

Minor points

1. The beginning of the results reads as more introduction.

In accordance with this comment, we have moved the beginning of the results reads to the introduction (final paragraph).

2. The MEFs with p53 KO are listed as 'described elsewhere' with no reference in the methods.

This was our mistake and we have quoted the reference (line 329).

3. The graphs of cell growth (3C, 3D) do not indicate the y-axis value in detail - relative cell number is percent of initial?

This is our lack of explanation, and we have added the following description into the legend of Figs. 3c and d: “The relative cell number indicates the total cell number/Total cell number at time 0 of each cell”.

4. There are no statistics on graph 3K.

As shown in Fig. 3i (original Fig. 3K), we have added statistical data.

5. The micrographs in Fig. 3F and 3H appear very different, while the data on the graphs are similar.

Micrographs give different impressions depending on the field of view. Therefore, only statistically certain results are shown, and the micrographs have been removed.

6. The pLKO.1 vector encodes shRNAs with a stem-loop structures rather than siRNA oligos.

Following to this comment, we have changed all vector names to shXXX.

7. SOX2 is a lineage transcription factor and may indicate a switch towards more squamous fate rather than stemness. Given SOX2 and OCT4 were not assayed, their mention appears tangential.

Following to this comment, we have deleted “,such as SOX2 and OCT4” from the following description: “In this context, STAT3 can help generate CSCs by inducing the expression of reprogramming factors, such as SOX2 and OCT4.” (lines 62 to 63)

8. Line 373 should read 'Trypan Blue exclusion'

We have changed (line 393).

Reviewer #2 (Remarks to the Author):

Sano and colleagues investigated the interplay between PRMT5 and STAT3. By using a series of cellular and biochemical assays they conclude that STAT3 regulates PRMT5 expression. Additionally, PRMT5 methylates STAT3, a post-transcriptional modification that is needed for full activation of STAT3 and for its tumorigenic potential.

Major points

1) Line 101: EGFR is not the most often mutated oncogene in NSCLC.

This was our mistake and we have changed this description as follows: “The one of the major oncogenic drivers in NSCLC is mutated epidermal growth factor receptor (EGFR) that result in constitutive activation.” (lines 95 to 97)

2) Across the whole manuscript and figures, it is not clear if the Authors are using siRNA of shRNAs. From the methods, it looks that they are using shRNAs expressed from lentivirus, but across the whole paper they use the expression siXXXX. Also sometimes, they refer to a “knock-down” when using the Crispr/Cas9 system (most likely they mean “knockout”). The authors mix the terms knockout and knockdown across the figures and test. This make difficult to interpret the results.

We agree with this comment. Following to this, we have changed all siXXX notations to shXXX. Furthermore, we have carefully changed the knockout and knockdown notations to the correct ones.

3) Across the manuscript different experiment are done in difference cell lines. There is not consistency which makes difficult to interpret the general picture.

We agree with this comment. We had analyzed the regulation of PRMT5-STAT3 in normal diploid cells and in NSCLC cells, but there were analyses that we did not do in NSCLC cells. So, we have performed additional experiments using NSCLC cells as shown in Fig. 2a, 2b, and 2f-g.

4) Figure 2: Key findings should be confirmed using a STAT3 knockout cancer cell line.

Specifically, the regulation of PRMT5, at mRNA and protein levels, by STAT3 upon stimulation with IL6 and EGF should be confirmed using STAT3 knockout cells.

According to this comment, we have performed additional experiment. The results are shown in Fig. 2A. The results have been described in the text as written in our response to the Reviewer #1 Point 3, as follows. According to this comment, we analyzed the induction of PRMT5 expression by STAT3 in NSCLC cells using STAT3 KO A549 and HCC827 cells. As shown in Fig. 2a and b, the induction of PRMT5 expression by IL-6 was markedly suppressed by STAT3 KO. Furthermore, we consider that Western blot loading to be even. From these results we have added the following description into the text: “To further analyse the induction of PRMT5 expression by STAT3 in NSCLC cells, we generated STAT3-knockout (STAT3-KO) cells by using the CRISPR/Cas9 system in A549 (EGFR wild type, adenocarcinoma) and HCC827 cells (EGFR mutant, adenocarcinoma). As expected, the induction of PRMT5 mRNA expression by IL-6 was markedly suppressed by STAT3 KO in A549 and HCC827 cells (Fig. 2a). Moreover, PRMT5 protein expression and induction by IL-6 were also suppressed by STAT3 KO (Fig. 2b), suggesting that PRMT5 is induced by STAT3 in NSCLC” (lines 156 to 162).

5) Figure 4C: It is difficult to interpret. Knockdown of PRMT5 results in downregulation of total STAT3 levels, which makes it difficult to conclude that there is less STAT3 activation (pY705). In addition, in the control cell line upon stimulation of IL6, there is more total STAT3. Thus, it is difficult to know if the increase of pY705 is due to IL6 or just to more total STAT3. A better WB has to be shown here.

Concerns about this result are also made in Reviewer #1 Point 4. Therefore, to make this result more definitive, we retried this experiment with different STAT3 antibody. As shown in Fig. 4c, the increase in the amount of STAT3-pY705 normalized to total STAT3 protein level by IL-6 was suppressed by shPRMT5. We believe that our results suggest that tyrosine phosphorylation of STAT3 is upregulated by PRMT5.

6) Figure 4f. M-STAT3 does not detect methylated STAT3, but just a methylated protein that is immunoprecipitated with PRMT5 and has around 80 kDa.

7) In general, the direct evidence that PRMT5 methylates STAT3 is quite weak. Figure 5c, as above, M-STAT3 does not detect methylated STAT3, but just a methylated protein of around 80 kDa (indeed, M-STAT3 seems to be present in the STAT3-KO cells). Previous

reports have shown that while PRMT5 may interact with STAT3, in vitro PRMT5 does not methylates STAT3 (Tee et al. Ref 63). Methylation levels of endogenous STAT3 in cell lines using mass spectrometry in presence and absence of PRMT5 should be shown. This is a key experiment on this the paper.

To address this concern, we performed the in vitro methylation assay. As a result, Myc-tagged wild type STAT3 (STAT3 WT purified by immunoprecipitation was methylated by recombinant human Flag-tagged PRMT5/His-tagged MEP50 complex (Fig. 4g, h). In contrast, methylation was clearly suppressed by arginine 609 (R609K) mutation and weakly suppressed by arginine 518 (R518K) mutation. From these results and the result of Fig. 4f (now Fig. 4i), we added the following description into the text: “To analyse whether STAT3 protein is methylated by PRMT5, we performed an *in vitro* methylation assay. Myc-tagged wild-type STAT3 (STAT3 WT) and its mutants purified by immunoprecipitation were methylated by recombinant human Flag-tagged PRMT5/His-tagged MEP50 complex (Fig. 4g, h). In contrast, upon amino acid substitution (arginine to lysine) at candidate methylation sites of STAT3, methylation was clearly suppressed by arginine 609 (R609K) mutation and weakly suppressed by arginine 518 (R518K) mutation (Fig. 4g, h). Next, wild-type STAT3 and STAT3 mutants were transiently expressed in the STAT3-KO normal human bronchial epithelial cell line BEAS2B (Supplementary Fig. 4h). As shown in Fig. 4i, methylated proteins could be detected at the same position as STAT3, and the degree of methylation was similar to that in the *in vitro* methylation assay. In addition, R609K mutant STAT3 lost most of its transcriptional activity in STAT3-KO BEAS2B cells (Fig. 5j). We also found that PRMT5 forms a complex with STAT3 via MEP50 (Fig. 4k). Furthermore, we showed that STAT3 and PRMT5 were colocalized in the cytoplasm, but only STAT3 was localized in the nucleus (Supplementary Fig. 4i, j). These results suggest that STAT3 is methylated in the methylosome complex formed by PRMT5/MEP50^{41, 42} in the cytoplasm, resulting in STAT3 activation, translocation into the nucleus, and target gene activation” (lines 204 to 221).

8) Figure 5c STAT3 and pY705 are detected in STAT3 knout cells.

To address this concern, we carefully performed the experiment with a different antibody. As a result, the non-specific bands were no longer visible. We believe that this result (now Fig. 4i) resolves this concern.

9) Figure 5e (showing STAT3 levels) is confusing. Parental cell line expressed only STAT3alpha?, “Vector” (STAT3 knock out) still expressed STAT3alpha?, wt and mutant reconstituted cell lines seems to express STAT3 alpha and beta. Or, is the lower band the myc-tagged version? Which usually it should run higher than non myc tagged STAT3? Nevertheless, STAT3 KO cells should not express STAT3. Or it is a knock-down.? In the figure legend is written both, Knockout and Knockdown. M-STAT3 does not really detect methylated STAT3.

As mentioned in the Point 8, the STAT3 band is no longer seen in STAT3 KO cells using different antibody, but the lower band was still seen in Myc-STAT3 expressing cells. Since this band is not seen with the Myc-tag antibody, suggesting that the N-terminal part containing the tag was specifically degraded for Myc-STAT3. From these results, we have added the following description into the legend of Fig. 5a: “The lower band of STAT3 was seen in Myc-STAT3 expressing cells. Since this band is not seen with the Myc-tag antibody, suggesting that the N-terminal part containing the tag was specifically degraded for Myc-STAT3.”

10) Assuming the cell lines described in figure 5e are used in the xenograft experiments, it is somehow surprising to see such big differences in tumor growth between the parental and the STAT3 KO (or knock-down?) cells, while the levels of STAT3 in both cell lines are not so different.

As described in the Point 8, 9, the STAT3 band is no longer seen in STAT3 KO cells using different antibody. Therefore, we believe that the difference in tumor growth is due to the loss of STAT3.

11) The co-localization experiments between PRMT5 and STAT3 are difficult to interpret. High resolution, high magnification pictures done with a confocal microscope should be shown. This should be sustained with a Duolink assay or similar.

In response to this concern, we performed immunofluorescent images of PRMT5 and STAT3 localization in A549 and HCC827 cells (Supplementary Fig. 4i). We believe these results suggested co-localization of PRMT5 and STAT3 in cytoplasm.

12) The STAT3 mutant R609K seems to be less phosphorylated (Y705), is it still able to translocate to the nucleus and bind DNA upon IL6 stimulation (compared to wt STAT3)?

In accordance to this comment, we performed additional experiments. As shown in Fig. 5b, R609K did not translocate to the nucleus in response to IL-6.

13) Across the manuscript the authors compare in several experiments more than two groups. A t-test is not appropriated

In experiments with more than two groups, Turkey's honestly significant difference test was applied for statistical comparisons. In Figures 2f, 2h, 3h, 5c, and Supplementary Figure 5f, the unpaired two-tailed Student's t-test was utilized.

Minor points

1) Figure 3m, n seems not to be in the manuscript, or refers to figure 3K i.

This is our mistake and Figure 3m,n was not shown. We have rewritten it to be accurate in the revised version.

2) Sometimes, the figures legends lack of information, such incubation times in the treatments

We have rewritten the figure legend to disclose sufficient information. We have included the incubation time, the number of cells, and the manufacturer and catalog number of the antibodies used in the Figure legend.

REVIEWERS' COMMENTS:

Reviewer #1 (Remarks to the Author):

The authors addressed my major concerns. The western blot in figure 2 with the STAT3 KO cells is more convincing than the original, which is now Supp Fig. 2. The authors did not repeat the ChIP-PCR but rather show data from ENCODE. The proper citations for ENCODE, cBioportal and CCLE (now DepMap) should be added.

Reviewer #2 (Remarks to the Author):

The authors have performed a very good revision and addressed most of my concerns. The manuscript has improved and I am happy to recommend its publication.

Minor points

Suggestion: Across the manuscript the authors claim that methylation of STAT3 at arginine 609 by PRMT5 is important for its transcriptional activity. However, the new data of figure 5b suggest that STAT3 R609K mutant is unable to translocate to the nucleus. Additionally, it seems that this mutant cannot be tyrosine phosphorylated (Fig 4J). Therefore, I would say that that R609 methylation is critical for STAT3 nuclear translocation/Y705 phosphorylation, rather than for STAT3 transcriptional activity. Since the mutation impairs nuclear translocation, it seems to be logical that the transcriptional activity is indirectly reduced. I suggest that the authors take this in account across the manuscript.

In some of the figures (e.g. figure 3) the authors sill use "siControl" and "shXXXX", I guess it should be shControl. Please, correct it

Line 215/216 Figure 5J should be 4J, please correct it

Please indicate the incubation time and cytokine concentration of the treatments in the figure legends, in some of them it is missing. E.g. Figure 5b IL6 treatment.

Responses to the reviewers' comments

MS: COMMSBIO-23-2230A

Title: PRMT5-mediated methylation of STAT3 is required for lung cancer stem cell maintenance and tumour growth

Authors: Yoshinori Abe, Takumi Sano, Naoki Otsuka, Masashi Ogawa, and Nobuyuki Tanaka

We are grateful to the reviewers for their invaluable comments and suggestions. In accordance with their suggestions, we adequately address the issues raised and have amended the paper accordingly. Please find below our point-by-point responses to each of the comments.

Reviewer #1 (Remarks to the Author):

The authors addressed my major concerns. The western blot in figure 2 with the STAT3 KO cells is more convincing than the original, which is now Supp Fig. 2. The authors did not repeat the CHIP-PCR but rather show data from ENCODE. The proper citations for ENCODE, cBioportal and CCLE (now DepMap) should be added.

In accordance with this comment, we have added the following information to the **Data availability** section as follows: “Data for PRMT5 or MEP50 expression, PRMT5 copy number and overall survival (OS) in lung adenocarcinoma was obtained from the cohort of GDC TCGA Lung Adenocarcinoma (15 datasets:

[https://xenabrowser.net/datapages/?cohort=GDC%20TCGA%20Lung%20Adenocarcinoma%20\(LUAD\)&removeHub=https%3A%2F%2Fxcna.treehouse.gi.ucsc.edu%3A443](https://xenabrowser.net/datapages/?cohort=GDC%20TCGA%20Lung%20Adenocarcinoma%20(LUAD)&removeHub=https%3A%2F%2Fxcna.treehouse.gi.ucsc.edu%3A443)). PRMT5

expression or lung squamous cell carcinoma was obtained from the cohort of GDC TCGA Lung Squamous Cell Carcinoma (15 datasets:

[https://xenabrowser.net/datapages/?cohort=GDC%20TCGA%20Lung%20Squamous%20Cell%20Carcinoma%20\(LUSC\)&removeHub=https%3A%2F%2Fxcna.treehouse.gi.ucsc.edu%3A443](https://xenabrowser.net/datapages/?cohort=GDC%20TCGA%20Lung%20Squamous%20Cell%20Carcinoma%20(LUSC)&removeHub=https%3A%2F%2Fxcna.treehouse.gi.ucsc.edu%3A443)

). In Figure 1c, protein levels of PRMT5 and STAT3 were obtained from Proteomics data of PRMT5 (O14744) and STAT3 (P40763) at the depmap portal. In supplementary Figure 1g, *PRMT5* expression level was obtained from expression public 23Q4 at the depmap portal. In supplementary Figure 1g, *PRMT5* expression data across cell lines was obtained from CCLE RNAseq gene expression data for 1019 cell lines at Cancer Cell Line Encyclopedia (CCLE). In

Supplementary Figure 2d, ChIP-seq data of STAT3 or H3K27Ac in H358 cells was obtained from Gene Expression Omnibus (GEO) at NIH National Center for Biotechnology Information (NCBI) (STAT3 ChIP-seq data: GSM2752894, H3K27Ac ChIP-seq data: GSM2752895)." (lines 517-533).

Reviewer #2 (Remarks to the Author):

The authors have performed a very good revision and addressed most of my concerns. The manuscript has improved and I am happy to recommend its publication.

Minor points

Suggestion: Across the manuscript the authors claim that methylation of STAT3 at arginine 609 by PRMT5 is important for its transcriptional activity. However, the new data of figure 5b suggest that STAT3 R609K mutant is unable to translocate to the nucleus. Additionally, it seems that this mutant cannot be tyrosine phosphorylated (Fig 4J). Therefore, I would say that that R609 methylation is critical for STAT3 nuclear translocation/Y705 phosphorylation, rather than for STAT3 transcriptional activity. Since the mutation impairs nuclear translocation, it seems to be logical that the transcriptional activity is indirectly reduced. I suggest that the authors take this in account across the manuscript.

In accordance with this comment, we revised the statement "Methylation of STAT3 at 609 by PRMT5 is important for priming for phosphorylation of tyrosine at 705, which is necessary for its nuclear transition" (Lines 25, 198, 272, 294). Furthermore, the schematic illustration in Figure 5h was adjusted to better depict the role of PRMT5-mediated STAT3 methylation. Moreover, we have changed "In this study, we found that PRMT5 can activate STAT3 via arginine methylation" to "In this study, we found that PRMT5 activates STAT3 via induction of tyrosine phosphorylation and nuclear translocation" in the Discussion (lines 272 to 273).

In some of the figures (e.g. figure 3) the authors sill use "siControl" and "shXXXX", I guess it should be shControl. Please, correct it

We have corrected siXXX to shXXX. (Figure 3B, Figure 4, Supplementary Figures 4 and 5, Supplementary table 2).

Line 215/216 Figure 5J should be 4J, please correct it

We have corrected Figure 5j to 4j in Line 215-216.

Please indicate the incubation time and cytokine concentration of the treatments in the figure legends, in some of them it is missing. E.g. Figure 5b Il6 treatment.

We have added a description of the cytokine concentration and treatment period in the manuscript (Line 812, 870, 920–921, and supplementary Figures).